# Implications of pre-diagnosis costs incurred by patients and their families for tuberculosis-related health-seeking behaviors in Mbeya and Songwe regions, Tanzania

Stella Kilima[1]*, Kamban Hirasen[2], Godfrey Mubyazi[1], Aneesa Moolla[2,3], Nyanda Ntinginya[1], Issa Sabi[1], Simeon Mwanyonga[1], Denise Evans[2]

1 National Institute for Medical Research (NIMR), Dar es Salaam, Tanzania, 2 Health Economics and Epidemiology Research Office, Faculty of Health Sciences, University of the Witwatersrand, Johannesburg, South Africa, 3 The Independent Institute of Education, IIEMSA, Ruimsig, South Africa

* stellakilima@yahoo.com, stella.kilima@nimr.or.tz

## Abstract

### Background

Pulmonary tuberculosis (PTB), remains one of the leading causes of death from infectious diseases worldwide, its burden is disproportionately affecting low- and middle-income countries (LMICs), including Tanzania. This study aimed to examine the direct and indirect costs incurred by patients and their families prior to PTB diagnosis, and how such costs influence patients' health-seeking behaviours.

### Methods

A mixed-methods approach was employed, integrating qualitative and quantitative methods for data collection, analysis, and interpretation. The study was part of a longitudinal observational cohort that included 261 adults diagnosed with PTB. Quantitative data were collected using a structured questionnaire adapted from the WHO's generic patient cost survey tool, administered after TB diagnosis was confirmed. To gain deeper insights, additional data were gathered through, in-depth interviews (IDIs) with purposively selected participants, aimed at identifying both financial and non-financial challenges experienced by patients and their family supporters before to diagnosis.

### Results

Of the 261 respondents, 59% were men, 48% were married, and 51% were living with HIV, with a median age of 35 years. The majority 185(72.8%) delaying visiting a healthcare facility (HCF) by more than four weeks following the onset of symptoms. The average total pre-treatment cost per patient was USD 28.8, comprising direct

**Data availability statement:** The data underlying this study are jointly owned by the study sites and the National Institute for Medical Research (NIMR) in Tanzania (reference: NIMR/ HQ/R.8a/Vol. IX/3527). Data access is governed by the Human Research Ethics Committee (HREC) of the University of Witwatersrand, Johannesburg, South Africa and the Mbeya Medical Research and Ethics Review Committee in Tanzania. All relevant data supporting the findings of this study are included within the paper, figures, and tables. The full datasets are available from the Health Economics and Epidemiology Research Office (HE²RO) for researchers who meet the criteria for accessing confidential data and have obtained the necessary approvals from the data owners. Requests for data access can be directed to HE²RO via email at information@heroza.org).

**Funding:** This work was supported by the German Ministry for Education and Research (GMBF) through the TB Sequel project under grant [LMU-IMPH-TB Sequel-01]. The funder had no role in the study design, data collection and analysis, decision to publish, or preparation of the manuscript.

**Competing interests:** The authors have declared that no competing interests exist.

medical, direct non-medical, and indirect costs. These costs varied based on the number and type of healthcare providers consulted. Qualitative findings indicated that participants frequently associated this pre-treatment costs with delays in seeking formal care, often influenced by self-medication, limited awareness of TB symptoms, and long distances to formal HCFs.

## Conclusion

This study highlights the significant financial and non-financial barriers faces by patients diagnosed with PTB and their families before diagnosis. Delays in care-seeking not only worsen health outcomes but also increase the overall economic burden. Addressing these challenges requires a comprehensive, multisectoral approach to improve early diagnosis, reduce patients costs and strengthen TB control efforts in high-burden settings.

## Background

### Overview of evidence on the economic burden of PTB

Pulmonary TB remains a major global public health concern. According to the World Health Organization (WHO), TB is among the leading causes of death from infectious diseases, with the majority of cases occurring in Low- and Middle-Income Countries (LMICs), where the burden is disproportionately high [1]. Each year, approximately 10 million people develop TB, and more than one million dies from the disease [1,2]. TB contributes significantly to healthcare utilization, including outpatient visits and hospitalisation, and continues to be a major driver of morbidity and mortality in affected populations.

### Economic burden of TB and pre-diagnosis costs

Beyond its clinical impact, TB imposes a considerable economic burden, particularly among socio-economically disadvantaged populations [3,4]. Although TB diagnosis and treatment are provided free of charge in many public health systems, patients often incur substantial out-of-pocket expenses, especially during the pre-diagnosis phase of care [5,6].These costs include direct medical expenses (consultation fees, tests, medications), direct non-medical costs (transportation, food, accommodation), and indirect costs (lost income due to time spent seeking care) [7–9]. Indirect costs are frequently excluded from traditional costs of illness studies, despite their substantial contribution to the overall financial burden [6,10]. These expenditures are often incurred across multiple providers, including pharmacies, private practitioners, and traditional or spiritual healers, before a confirmed TB diagnosis is made in the formal health system.

Evidence consistently shows that pre-diagnosis costs are critical component of the overall economic burden of TB, particularly in high burden settings [11]. A systematic review of the economic burden of TB in Africa found that average pre-diagnosis costs

were $36 in Malawi, $109 in Zambia, and $196 in Ethiopia accounting for 11.3%, 10.6%, and 35.0% of pre-diagnosis household income, respectively [11]. In The Gambia, a recent study reported average pre-diagnosis direct costs of $22.93, constituting nearly 47% of the total direct costs incurred throughout the TB care continuum [12]. This phase represented the highest income loss, with patients who consulted traditional healers incurring the greatest average expenses of $30.80 and spending up to 50 hours seeking care, substantially more than those who visited private clinics, who reported average costs of $14.84 and only 5 hours of care seeking time [12].

## Catastrophic costs and opportunity losses

Pre-treatment expenditures, though often overlooked, can be financially devastating for TB-affected households [5,13]. These out-of-pocket (OOP) costs can delay diagnosis, reduce access to treatment, and negatively impact treatment adherence and outcomes, ultimately reinforcing the cycle of poverty and poor health. TB-related illness often leads to reduced productivity, absenteeism and diminished earning potential [6,10].

According to the WHO definition, TB-related "catastrophic costs" occur when total costs of TB care seeking including direct medical, direct non-medical and indirect costs exceed 20% of a household's annual income or expenditures [14]. These financial hardships reduce the ability of affected families to meet basic needs and may deter care-seeking, particularly when upfront payments are required [15,16]. Geographic and systematic barriers further exacerbate these issues, especially in rural areas where transportation costs and opportunity losses are substantial [17,18].

## TB services in Tanzania

Tanzania is one of the 30 high TB burden countries globally, yet it has made notable progress in recent years. According to the National TB and Leprosy Programme (NTLP 2023), the national TB incidence rate declined from 306 per 100,000 population in 2015–183 per 100,000 in 2023 representing a reduction of 40% [19]. TB-related mortality also dropped by 68%, from 58,000 deaths in 2023–18,400 in 2025 [19]. Furthermore, case notification rates increased by 6% annually between 2015 and 2022, peaking at 165 per 100,000 population, but showed a slight decline in 2023, which is 145 per 100,000 population [19]. To strengthen TB control efforts, the Tanzanian government, in collaboration with international partners, has expanded TB services through decentralization and integration into lower-level health facilities [20]. However, limited access to diagnostic services remains a challenge [21,22]. Only 45% of the 655 health care facilities (HCFs) in the Mbeya and Songwe regions offer TB care, and just 99 facilities (15%) are accredited for TB diagnostics [23]. Advanced diagnostic tools such as GeneXpert and drug susceptibility testing (DST) are largely confined to regional hospitals, limiting access in rural settings. Lower-level facilities often rely on clinical diagnosis and sputum smear microscopy, which have limited sensitivity, particularly in individuals living with HIV and extrapulmonary TB. Despite existing sample referral systems (from lower to higher-levels facility), logistical delays persist [24]. The TB services are offered mainly through public and a few selected private health facilities. In Tanzania, government health facilities serve about 72% of TB patients for diagnosis and treatment [19].

Limited access to accurate diagnostic tools, combined with low public awareness of TB symptoms, contributes to delayed diagnosis and extended care-seeking pathways [25]. Patients commonly consult multiple providers before obtaining a confirmed diagnosis, resulting in increased OOP expenses [25]. These delays elevate the financial burden, compromise treatment outcomes, and heighten the risk of continued TB transmission [25].

In regions such as Mbeya and Songwe, this pattern is particularly evident. Patients often seek care at multiple facilities or non-formal providers before receiving an accurate diagnosis. This care-seeking trajectory leads to high pre-treatment costs that many households can ill afford, making them vulnerable to catastrophic financial consequences even before initiating treatment.

### Importance of addressing pre-diagnosis costs in Tanzania

Despite their significance, pre-treatment costs are often excluded from cost-of-illness studies, leading to an incomplete picture of TB's economic burden. Recognizing and quantifying these early-stage financial burdens is essential for informing health policies and designing interventions aimed at reducing economic barriers to care. Comprehensive inclusion of pre-treatment costs in research and policy frameworks is critical for achieving universal health coverage, improving treatment outcomes, and breaking the cycle of poverty associated with TB. This study examines the economic burden associated with the pre-diagnosis phase of pulmonary TB care, emphasizing patient and household experiences within high-burden contexts in Tanzania. By highlighting this under-researched aspect of the TB care continuum, the findings aim to inform equitable and cost-sensitive strategies for TB control.

## Materials and methods

### Study design

This study, employed a mixed-methods approach, incorporating both qualitative and quantitative components, and was nested within a longitudinal, multi-center observational cohort study known as TB Sequel [26]. TB Sequel 1(2016–2022) was a longitudinal cohort study aimed at understanding the long-term clinical, socio-economic and respiratory health outcome among individuals diagnosed with pulmonary TB. This parent study was conducted across four Sub-Saharan Africa (SSA) countries: Mozambique, Tanzania, The Gambia, and South Africa. However, this paper focuses exclusively on data collected from participants enrolled in Tanzania.

### Setting

The study was conducted in two study regions, Mbeya and Songwe, located in the southwestern highlands of the United Republic of Tanzania (URT), specifically on the mainland. The TB Sequel project recruited newly diagnosed TB patients from the 24 randomly selected HCFs across the study regions. The HCFs were selected among those officially accredited to provide both TB diagnosis and treatment services, including 13 hospitals, 10 health centres and one dispensary. These HCFs were selected to refer newly diagnosed TB patients to the TB Sequel project, which is hosted at Mbeya Zonal Referral Hospital (MZRH) in the City of Mbeya. Interviews were conducted in a quiet, private area within MZRH, separate from the TB Sequel clinic. It is important to note that while participants took part in the TB Sequel study, they continued to receive their standard TB treatment at the referring facilities in line with national guidelines and the standard of care. The TB Sequel study activities were designed to run in parallel with routine care, without interfering with clinical management.

### Study population

Convenience sampling was employed, whereby eligible patients presenting at participating health care facilities were referred to the study as they became available.

Referred patients underwent confirmatory tests at the TB Sequel clinic before enrolment. Recruitment occurred following a TB diagnosis, with only individuals confirmed as sputum-positive enrolled in the study. The recruitment period spanned approximately two years. TB Sequel enrolled participants only at treatment initiation; therefore it did not capture or intervene in the pre-diagnostic patient journey and instead asked patients to recall their costs from the onset of TB symptoms until diagnosis.

Participants benefited from additional diagnostic services provided free of charge by the TB sequel project, beyond what is typically offered in routine care. These services included spirometry, electrocardiogram (ECG), 6-minute walk test (6MWT), chest X-ray, and urinalysis. Moreover, follow-up visits were conducted for individuals who had completed their TB treatment, offering continued monitoring and care as an aspect which was highly valued by participants. To further

reduce participation barriers, transport costs were reimbursed, which helped alleviate financial constraints. These combined efforts significantly contributed to enhancing participant engagement and retention in the project.

The inclusion criteria for participants in the **quantitative** component were: A positive result for *Mycobacterium tuberculosis* (MTB) from at least one sputum sample, as determined by Xpert MTB/RIF assay in the study clinic/laboratory, or a positive culture result from either the study laboratory or other accredited TB laboratory [26,27]; aged 18 years or older; willing to provide a written informed consent (IC), or witnessed oral consent if illiterate before the collection of the first sample or any other study-related data; willing to undergo HIV testing; consent to provide and allow collection and storage of blood, urine, and sputum samples; Willing to initiate anti-TB treatment following diagnosis. Lastly, residence within the study area and willingness to notify the study team of any address changes during the treatment and follow-up [26]. Exclusion criteria included: receipt of anti-TB treatment within the past 6 months; inability to produce and provide two sputum samples; severe medical or psychiatric conditions (as determined by the site investigator) that could impair the ability to provide informed consent or comply with study requirements; participation in ongoing investigational trials related to TB or lung diseases and lastly currently imprisoned.

For the **qualitative study** component, participants were drawn from the ongoing TB Sequel project. A purposive sampling strategy was employed to select individuals attending routine clinic visits. Eligible participants were identified and provided with detailed information about the qualitative study. To minimize financial burden, interviews were scheduled to coincide with routine follow-up appointments.

Eligibility was determined based on predefined inclusion and exclusion criteria. To be eligible, participants were required to have a confirmed diagnosis of PTB and be currently receiving treatment or attending a follow up visit. Additional inclusion criteria included being 18 years of age or older, the ability to communicate comfortably in Kiswahili or English, and willingness to participate in a face-to-face interview and provide written informed consent, or witnessed oral consent if illiterate. Individuals were excluded if they had completed TB treatment and all follow up visits, were under the age of 18, had significant medical or mental health conditions (as determined by the site investigator) that could affect their ability to provide informed consent or had previously interrupted TB treatment and subsequently restarted therapy.

## Data collection

**Quantitative surveys.** Trained researchers conducted structured interviews using a questionnaire adopted from the WHO's general patient cost survey [15]. The original English version of the questionnaires was verbally translated into the local language by experienced researchers fluent in both English and the local language (Kiswahili). A standardized structured questionnaire was maintained across all participating countries to ensure consistency. Before data collection, the research team, including research assistants, underwent a five-day training to familiarise themselves with the study objectives, methodology, and ethical considerations. This was followed by a pilot test of the questionnaire. Based on the pilot feedback, the questions were revised to enhance clarity, user-friendliness, and contextual relevance, ensuring questions were easily understood by the study participants [28]. Data collection was paper-based. Completed questionnaires were collected weekly and submitted to a Data Clerk for double entry into an OpenClinica software. The dataset was subsequently imported into SAS version 9.4 for analysis. Patients' pre-treatment costs were calculated by assessing expenses incurred from the onset of PTB symptoms until the time of diagnosis. The analysis focused on direct medical costs, direct non-medical costs, and indirect costs. Time expenditure was used as a proxy for opportunity cost, capturing potential income loss due to hours spent without working [8].

**Qualitative interviews.** An IDI guide was developed, pretested, and used to facilitate qualitative data collection. The guide included questions on basic demographic characteristics, such as education level, age, and marital status, as well as the history and duration of TB symptoms before seeking care. It also explored the costs associated with all healthcare providers visited, participants' perception of the availability and quality of health facilities, and related OOP expenses. The original English versions of the qualitative interview guide was fully translated into written Kiswahili, and

the translation was independently reviewed for accuracy and consistency by two bilingual researchers fluent in both English and Kiswahili. After completing their clinic visit, individuals exiting the TB facility were approached by a member of the research team to determine their interest in participating in an in-depth interview (IDI). Those who agreed provided written IC; for those who were illiterate, oral consent was obtained and documented with a thumbprint. Participation in the IDI was separate from the main TB Sequel cohort study [29]. Separate written consent was obtained for audio recording of the interviews. Qualitative data were gathered using open-ended questions designed to complement and enrich the quantitative findings [27]. The interviews were conducted by two trained and experienced social scientists, a moderator, and a note-taker.

This study employed a pre-established interview guide developed for individuals diagnosed with TB. IDIs were conducted until thematic saturation was reached, indicating that no new information was emerging from subsequent interviews [30].

## Data analysis

**Quantitative data.** Quantitative data were analysed using SAS version 9.4 (SAS Institute, Cary, North Carolina, USA). The analysis aimed to describe participants characteristics, care-seeking behaviours and the costs associated with such behaviours. Therefore, the travel time, visit time (including waiting time), and costs in USD (direct medical, direct non-medical, indirect, and total) were stratified by the number of providers the participants reported to have visited before a PTB diagnosis and treatment initiation. Data were categorized to highlight averages, extremes, and key measures of central tendency, such as medians and interquartile ranges (IQR). Cost data are summarized using mean and standard deviation, consistent with standard practices in economic evaluation [31].

The nature of the data influenced the selection of appropriate analytical methods, depending on whether the variable was categorical or continuous in nature. The 'pre-treatment costs' in this paper referred to all costs incurred from the onset of TB symptoms until the initiation of TB treatment, as reported at baseline (Month 0 or the start of TB treatment). The analysis involved comparing costs incurred by respondents from different residential settings (urban versus rural), sex (males versus females), and marital status (married or cohabiting versus single individuals).

Using the human capital valuation approach [32], we used the product of total time lost due to TB (in hours) and multiplied it by the value of the hourly wage rate in Tanzanian shillings. For individuals with cash-based income, such as those working in formal employment sectors, we determined an hourly wage rate by dividing the monthly salary/wage received by the individual before a TB diagnosis by the total number of hours worked in one month. Accordingly, the monetary value of productivity time lost was estimated by multiplying the calculated hourly wage rate by the total number of working hours lost during the month. However, for those informally employed, such as domestic workers, street vendors, farmers, and the like whose working hours and monetary earnings were unspecified, we utilised an hourly rate that was equivalent to the minimum hourly wage of $0.29 in Tanzania for 2021 [33]. The costs reported before 2021 were first inflated to 2021, using the 2021 World Bank GDP deflator, and then converted to the USD using the average exchange rate (USD 1 = TZS 2297.76, based on currency converter http://www.oanda.com/currency/converter) [33].

Given the non-normal distribution of cost data, non-parametric tests were employed to evaluate differences in pre-treatment costs across groups. The Kruskal-Wallis test (¥) was used for comparisons involving more than two categories, while the Wilcoxon rank-sum test (§) was applied for binary comparisons. Statistical significance was defined as $p < 0.05$.

**Qualitative data.** Audio-recorded in IDIs were transcribed verbatim and translated into English by two experienced research assistants. Transcripts were independently reviewed against field notes for quality assurance and data validation [34]. The principal investigator (PI) verified transcript accuracy by cross-checking with the original audio recordings. All transcripts were imported into NVivo 12 for thematic analysis. A mixed deductive-inductive approach using axial coding was employed. A codebook was developed to ensure consistency in coding. Three trained analysts reviewed the transcripts to familiarize themselves with the content, followed by systematic coding. Themes and sub-themes were identified in line with study objectives. Discrepancies or new themes emerging during coding were resolved through discussions to achieve

consensus. Although the broader project identified three main themes and ten sub-themes, this paper focuses on two main themes and six sub-themes (Table 1). These are discussed in detail in the qualitative results section.

## Ethics considerations

The quantitative study protocol was reviewed and approved by the National Institute for Medical Research (NIMR) in Tanzania (Reference: NIMR/HQ/R.8a/Vol. IX/2449). Also, the qualitative study protocol was approved by the Heath Research Ethics Committee (HREC) of the University of the Witwatersrand, South Africa (M200570, 29 May 2020), the Mbeya Medical Research and Ethics Review Committee (16 January 2020), and more importantly, the National Institute for Medical Research in Tanzania (Reference: NIMR/HQ/R.8a/Vol. IX/3527). In this study, all participants consented to participate by signing the informed consent form.

## Inclusivity in global research

Additional information regarding the ethical, cultural, and scientific considerations specific to inclusivity in global research is included in the Supporting Information (S1 Checklist).

## Results

In the quantitative study, 319 participants were initially recruited. Of these, 49 did not meet the eligibility criteria, 6 tested negative for TB, and 3 declined to proceed with follow-up procedures. Consequently, 261 participants with laboratory-confirmed PTB, representing 81% of those initially recruited, were enrolled in the TB Sequel study. All enrolled participants completed the socio-economic questionnaire after their routine clinical visits.

## Background information

Participants were mainly between 25 and 34 years (Table 2). About two-thirds of participants (62.8%) had primary education and only 37.1% were employed, either formally or informally. Nearly half of the participants (47.9%) were married, and approximately 51% were diagnosed with PTB while also living with HIV. Almost all respondents (98.8%) were initiated on the TB standard regimen, two months of rifampicin (R), isoniazid (H), pyrazinamide (Z), and ethambutol (E) (RHZE), followed by four months of R and H, particularly for the drug-susceptible TB [35]. Four patients were started on a second-line anti-TB regimen because they were diagnosed with Multidrug-resistant tuberculosis (MDR-TB).

Table 1. Identified qualitative themes and sub-themes.

| Themes | Sub-themes |
| --- | --- |
| Healthcare seeking pathways, underlying causes, and financial burden before PTB diagnosis | Financial expenditures for pre-treatment care-seeking |
| | Delays in contacting formal healthcare facilities due to limited knowledge of PTB symptoms |
| | Coping strategies for managing PTB symptoms |
| | Influence of domestic and social obligations on delayed healthcare-seeking |
| Factors external to the formal health care system underlying a delayed PTB care-seeking | Rural-Urban differences in TB Care Accessibility |
| | Perceived quality of care related costs |
| | Limited knowledge of PTB symptoms |
| Experience before visiting a health facility and the cost associated | Informal support received before being TB diagnosis |
| | Referral and companion support received from family members and associated cost |
| | Feelings about pre-treatment procedures costs |

**Table 2. Total pre-treatment patient costs by patient characteristic (n = 261).**

| Characteristics | Description | Quantitative sample (N = 261) N (%) | Total pre-treatment cost, Median (Q1-Q3) | Significance test (Kruskal-Wallis test¥/ Wilcoxon rank-sum§) |
|---|---|---|---|---|
| Age (years) | Median (IQR) | 35.0 (29.0-43.0) | – | P-value (0.19) ¥ |
| | 18-24 | 31 (11.9%) | 13 (4.8-32.3) | |
| | 25-34 | 98 (37.7%) | 22.1 (10.7-40.8) | |
| | 35-44 | 75 (28.9%) | 14.8 (7.1-35.5) | |
| | ≥45 | 56 (21.5%) | 15.9 (3.3-34.2) | |
| Sex | Female | 108 (41.4%) | 15.3 (8.8-33) | P-value (0.46) § |
| | Male | 153 (58.6%) | 21 (7.2-37.4) | |
| Employment | Yes | 93 (37.1%) | 19.9 (10.2-38.4) | (0.21) § |
| | No | 158 (62.9%) | 18.9 (7.1-34.9) | |
| Education level | No schooling | 29 (11.1%) | 20.8 (10.3-32.5) | P-value (0.24) ¥ |
| | Primary school | 164 (62.8%) | 15.9 (7.1-38.2) | |
| | Secondary/high school | 57 (21.8%) | 26.4 (12.1-37.4) | |
| | Vocational training | 6 (2.3%) | 14.2 (3.7-21.4) | |
| | University or higher | 5 (1.9%) | 7.4 (1.2-24.6) | |
| Relationship status | Single/living with a partner | 64 (24.5%) | 17.3 (7.5-32.3) | P-value (0.01)¥* |
| | Married | 125 (47.9%) | 22.6 (9.7-42.1) | |
| | *Divorced/separated/widowed* | *72 (27.6%)* | *10.3 (6.7-25.2)* | |
| **HIV status** | Negative | 128 (49.0%) | 19.3 (9.6-40.2) | P-value (0.29) § |
| | Positive | 133 (51.0%) | 18.7 (6.4-34.2) | |
| **Smear microscopy** | Negative | 13 (5.0%) | 42.1 (5-50.7) | P-value (0.60) § |
| | Positive | 246 (95.0%) | 19 (8.6-36.3) | |
| **Treatment regimen** | 2HRZE/4HR | 257 (98.9%) | 19.1 (7.5-37.2) | P-value (0.81) § |
| | Other | 3 (1.2%) | 17.6 (9.3-47.1) | |

*IQR: inter-quartile range; HIV: human immunodeficiency virus; 2HRZE: 2 months of isoniazid, rifampicin, pyrazinamide, and ethambutol; 4HR: 4 months of isoniazid and rifampicin.*

*Significance test¶ (Kruskal-Wallis test¥/ Wilcoxon rank-sum§).*

Table 2 presents pre-treatment costs and stratified by several key patient characteristics at baseline (Month 0 or the start of TB treatment). Costs were similar across most baseline patient characteristics (p > 005). The analysis showed a significant difference in costs by marital status (p = 0.01), with married individuals incurring higher pre-treatment costs compared to those who were divorced, separated, or widowed (Table 2).

**Respondents' health care seeking behaviours pre-treatment initiation**

Fig 1 illustrates the pre-diagnosis health-seeking behaviour or pathways of individuals diagnosed with PTB. From the overall sample (n = 261), a third had not contacted any other service providers before presenting to the referral HCFs (where they were diagnosed). The remainder (69.3%) reported contacting one or several care providers before being diagnosed. Among those who reported contacting any provider, 50% indicated that they had contacted a retail pharmacy (91). Furthermore, it has been demonstrated that over 40% had visited a public hospital as their second or third provider (Fig 1).

Table 3 presents the duration between the onset of symptoms and the initiation of TB treatment. The median time to treatment initiation was 8 weeks, with an interquartile range of 4–16 weeks. Furthermore, only a small proportion of patients (1.2%) started treatment within one week of symptom onset, as the median time from sputum collection to

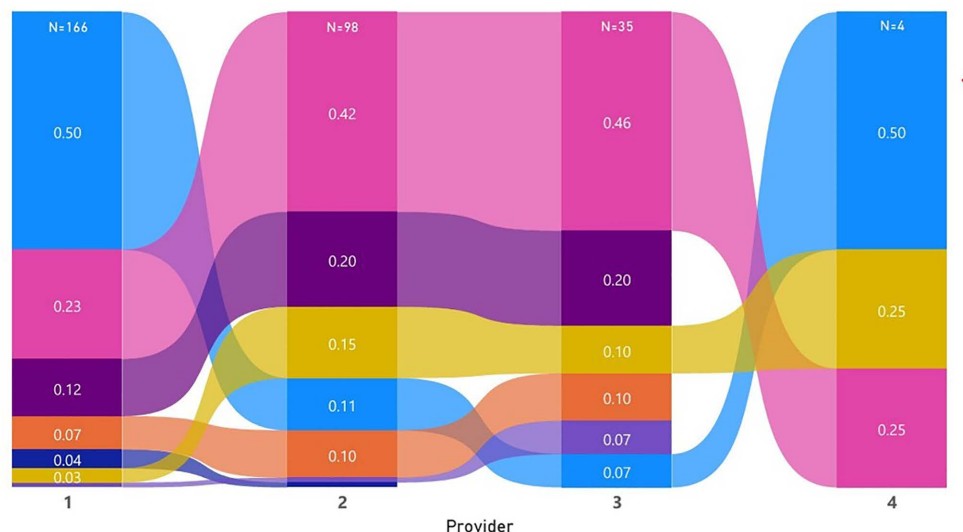

**Fig 1. Order and frequency of pre-treatment providers seen.**

**Table 3. Pre-treatment health-seeking behaviour and time from symptoms onset to treatment initiation (N=261).**

| Time from symptoms onset to DS-TB treatment initiation (weeks) | Median (IQR) | 8.0 (4.0-16.0) |
|---|---|---|
| | ≤ 1 | 3 (1.2%) |
| | 1-2 | 12 (4.7%) |
| | 2-4 | 54 (21.3%) |
| | ≥4 | 185 (72.8%) |
| **Pre-treatment health-care provider 1** | | |
| Travel time to provider (hours) | Median (IQR) | 1.5 (0.7-3.0) |
| Visit time at provider (hours) | Median (IQR) | 1.0 (0.5-4.0) |
| **Pre-treatment health-care provider 2** | | |
| Travel time to provider (hours) | Median (IQR) | 1.5 (1.0-2.8) |
| Visit time at provider (hours) | Median (IQR) | 4.0 (2.0-8.0) |
| **Pre-treatment health-care provider 3** | | |
| Travel time to provider (hours) | Median (IQR) | 2.0 (1.0-4.0) |
| Visit time at provider (hours) | Median (IQR) | 6.0 (2.5-14.0) |

*IQR: inter-quartile range.*

treatment initiation ranges from 2 to 7 days for drug-sensitive TB, even when molecular diagnostics are available. The majority of participants (72.8%) experienced a delay of more than four weeks before starting TB treatment. These findings highlight a significant delay in treatment initiation, which could contribute to ongoing transmission, worse health outcomes, and increased financial and social burdens (Table 3).

It was noted that visit duration tended to increase with each successive additional provider, suggesting a cumulative pattern of time spent during consultations. Travel time appears to behave additively, as the total distance travelled to the first provider is comparable to that of the third provider, while visit time accumulate across encounters (Table 3).

The costs that were reported for direct medical expenditures encompassed fees for medical consultations, screening through radiography or labs, and medications. These costs increased as the number of providers contacted increased. The average total direct medical costs were USD16.54, with costs for the first, second, and third providers being USD6.82, USD11.58, and USD12.04, respectively. Non-medical costs and indirect costs also varied with the number of contacted providers. It is important to note the variability in the cost data, which may be influenced by some extreme values (Table 4).

The average pre-treatment total cost (direct medical + direct non-medical + indirect costs) per patient was USD 28.8 (SD35.8). These costs varied according to the number and type of healthcare providers visited, increasing from USD9.95 (SD10.8) to USD19.1 (SD23.0), and USD22.0 (SD23.5) for one, two, and three or more providers that were visited, respectively (Table 4). The monthly national minimum wage in Tanzania is approximately USD 186 [36–38]. Notably, the cost of seeking care from a single provider (USD9.95) accounts for nearly 5.3% of this amount. For individuals who consulted multiple providers, the cumulative pre-treatment expenses could consume up to 15.5% of their monthly income. This financial burden is even more pronounced among individuals without employment or stable income who constitute a large proportion of participants in this study.

**Table 4. Costs in 2021 USD stratified by the number of providers visited.**

| Cost category | Description | Provider 1(n=166) | Provider 2(n=98) | Provider 3+(n35) | Total (mean, SD) |
|---|---|---|---|---|---|
| Direct medical costs (mean, SD) | Consultation | 0.6 (1.4) | 1.7 (4.1) | 1.6 (2.1) | 2.0 (3.8) |
| | Radiography/imaging | 0.2 (1.3) | 0.8 (2.4) | 1.3 (2.9) | 1.0 (2.7) |
| | Other procedures | 0.1 (0.9) | 0.2 (1.1) | 0.1 (0.8) | 0.2 (1.3) |
| | Laboratory tests | 0.6 (1.8) | 0.9 (1.9) | 1.0 (3.3) | 1.4(2.8) |
| | Drugs | 5.0 (6.0) | 7.4 (12.4) | 8.1 (13.9) | 11.2 (14.2) |
| | Other (supplementation) | 0.1 (0.5) | 0.5 (2.1) | 0.0 (0.1) | 0.4 (1.7) |
| | Hospitalization | 0.3 (2.3) | 0.3 (1.9) | 0.0 (0.0) | 0.5 (2.8) |
| | **Total direct medical costs** | **6.7 (8.1)** | **11.6 (13.7)** | **12.0 (14.5)** | **16.5 (16.5)** |
| Direct non-medical costs (mean, SD) | Transportation | 0.6 (1.0) | 1.8 (2.9) | 1.9 (2.5) | 2.1 (3.3) |
| | Food | 0.2 (0.5) | 0.5 (1.1) | 0.8 (0.9) | 0.7 (1.2) |
| | Accommodation | 0.0 (0.0) | 0.0 (0.0) | 0.1 (0.7) | 0.0 (0.4) |
| | Other – non-medical | 0.0 (0.4) | 0.2 (1.1) | 0.1 (0.3) | 0.2 (1.0) |
| | **Total direct non-medical costs** | **0.8 (1.4)** | **2.4 (3.8)** | **2.9 (3.5)** | **2.9 (4.3)** |
| **Total direct costs (mean, SD)** | **Direct medical + direct non-medical costs** | **7.5 (8.7)** | **13.6 (15.1)** | **14.6 (16.6)** | **19.3 (19.3)** |
| Indirect costs (mean, SD) | Patient travel time to pre-treatment providers | 1.0 (1.5) | 1.81 (5.5) | 1.9 (3.4) | 2.6 (5.8) |
| | Patient visit time at pre-treatment providers | 1.7 (3.7) | 3.6 (8.7) | 6.2 (10.8) | 5.4 (11.8) |
| | Total patient indirect costs (travel + visit time) | 2.7 (4.4) | 5.5 (12.5) | 8.1 (13.4) | 7. 9 (16.5) |
| | Family costs – transport | N/A | N/A | N/A | 6.56 (14.1) |
| | Family costs – food | N/A | N/A | N/A | 1.79 (3.3) |
| | Family costs – accommodation | N/A | N/A | N/A | 0.51 (2.4) |
| | Family costs – other | N/A | N/A | N/A | 0.34 (1.5) |
| | Total family costs | N/A | N/A | N/A | 8.78 (16.2) |
| Total indirect costs (mean, SD) | Patient + family costs | N/A | N/A | N/A | 10.09 (22.1) |
| Total pre-treatment costs (mean, SD) | Total direct + total indirect costs | 10.0 (10.8) | 19.1 (23.0) | 22.0 (23.3) | 28.8 (35.8) |

Abbreviations: *IQR: inter-quartile range; DS: drug-susceptible; TB: tuberculosis; SD standard deviation.*

**Healthcare-seeking pathways, underlying causes, financial burden before PTB diagnosis**

This theme explores participants' challenging experiences prior to receiving formal PTB diagnosis. Many reported incurring substantial financial costs when seeking treatment, often purchasing various medications and consulting multiple healthcare providers without knowing the underlaying causes of their symptoms. Several individuals described visiting informal and non-specialized providers before eventually accessing formal TB related services. A common cited reason for this pattern was the absence of nearby formal HCFs, which contributed to delays in receiving an accurate diagnosis and appropriate care.

> *I spent a lot of money on different medications to treat a persistent cough. Although the symptoms eased for a short while, the cough kept returning. I eventually resorted to stronger medication, unaware that it was actually TB. After about six months with no improvement, I decided travel to a government hospital located far from here for a tuberculosis test, which confirmed I was positive"* (Male, 35 years old, Mbeya region).

Participants expressed regret over the loss of productive time and financial resources incurred while traveling to distant providers. In the absence of the accessible formal care, many reported relying on self-medication from local drug shops and frequently consulting traditional healer, both of which were described as common practices. Qualitative respondents also highlighted a general lack of awareness regarding PTB symptoms and the costs associated with diagnostic services. Among those who were able to recognize the symptoms and understood their potential consequences, limited financial resources particularly for transportation and pre-diagnosis service emerged as primary barriers to timely and continuous PTB care-seeking. The following statements illustrate participants' experiences:

> *"It is common practice in our village to begin treatment by obtaining medicine from a local drug shop, especially since we have a well-stocked one nearby. However, some community members prefer first to consult traditional healers. If there's no improvement after using the prescribed medicine, the next step is to visit a hospital.* (Male, 48 years old, Tunduma district, Songwe region).

> *"Accessing health services can be challenging. For example, I struggled to find a nearby facility for TB diagnosis and had to travel to Kiwanja-Mpaka Health Center, which caused inconvenience due to transportation and financial costs."* (Female, 38 years old, Mbeya region).

Many participants reported that initially patients relied on self-medication, purchasing over-the-counter remedies such as cough syrup, pain relievers, or antibiotics like amoxicillin. However, when these treatments proved ineffective, participants often experienced frustration and a sense of financial loss. This experience heightened their awareness of the seriousness of their condition, ultimately prompting them to seek professional medical care. One participant shared experience of starting with cough syrup, then progressing to antibiotics, before realizing after three months that the illness was more serious than initially thought, leading to a hospital visit for a proper diagnosis

> *"When I first started coughing, I assumed it was just a normal cough, so I bought some cough syrup. However, it didn't help. As the condition worsened and I noticed no improvement, a pharmacist recommended an antibiotic. But by then, the cough had become much more severe, and I began to realize that this was not normal cough. It took about three months before I finally decided to visit the hospital for a more thorough diagnosis* (Female, 44 years old, Mbeya region).

The following pictorial diagram illustrates the patient pathways and the associated factors before tuberculosis treatment initiation. The diagram maps the sequential care-seeking steps taken by participants, showing how

various factors such as proximity to care, provider type, and delays influence the trajectory. This visual summary helps readers follow the multi-step decision-making process and identifies key determinants affecting care pathways (Fig 2)

## Financial expenditures for pre-treatment care-seeking

Participants reported that the onset of symptoms often compelled them to choose among the available care providers based primarily on proximity rather than perceived quality of care. Consequently, consulting traditional healers or practitioners was frequently viewed as the most accessible and immediate option, often complementing modern pharmaceuticals obtained from nearby retail drug outlets. However, the cost of traditional medicines prescribed by these traditional healers was often prohibitively high for less privileged individuals. Several respondents recounted personally paying OOP expenses or knowing someone who had paid money to compensate a traditional healer. For example, one woman confessed to having stayed with a traditional practitioner for approximately sixty days. During this period, her family provided food such as maize, flour, and beans to support her stay. Moreover, seeking care from a traditional healer across a national border was also reported as a common practice.

*"I stayed with the traditional healer for about two months, during which he treated me with a specific medicine that I drank daily. I spent nearly all my money there because he instructed us, as a family, to bring food to last the entire stay. This placed a significant financial burden on my family and me personally, as I had to take food meant for my household, leaving my children with very little for their own consumption"* (Female, 43 years old, Songwe region).

*"I was running from one traditional healer to another. I even went to Malawi, where I was told there is a traditional healer who is good at handling this cough problem. However, I did not get any relief while there, and instead, I ended up losing all my money on transport and procuring medicine. If I knew earlier that I was suffering from TB, how could I have dared to do that?"* (Male, 40 years old, Songwe region)

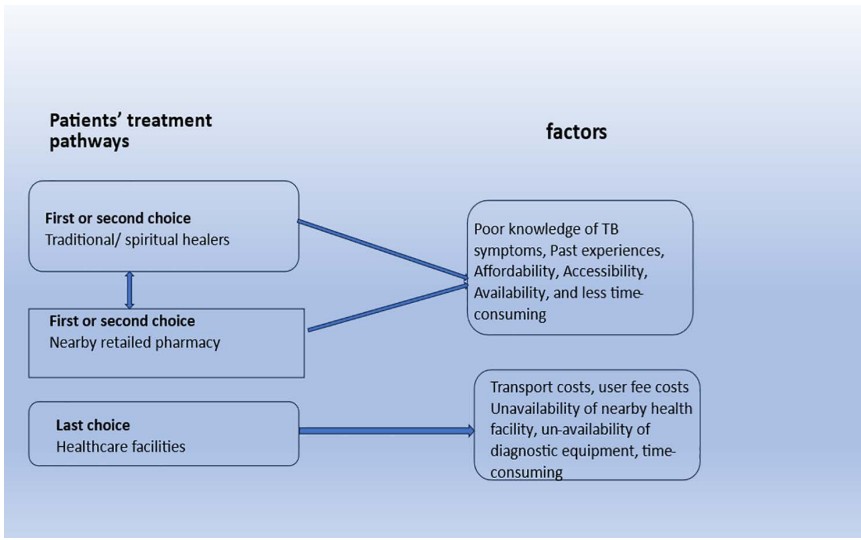

**Fig 2. Pre-treatment health seeking pathways and associated factors.**

**Delays in contacting formal healthcare facilities due to limited knowledge of PTB symptoms**

Delayed attendance at formal health facilities for PTB diagnosis and treatment was frequently reported as a regrettable factor contributing to unexpected high costs. This delay was particularly pronounced among individuals lacking prior knowledge of PTB symptoms. Interestingly, the comments from several respondents conveyed a shared sense of regret experienced by patients and their families, especially when delayed care led to referral to a distant health facility that imposed user fees. Such delays not only exacerbated financial burdens but also hindered timely access to appropriate care.

> "*I coughed for more than two months without realizing it was tuberculosis because I was unaware of the symptoms.*" (Male, 35 years old, Mbeya City, Mbeya Region).

> "*I spent over a year seeking treatment until a friend suggested I go to a government hospital for a diagnosis. That's when I found out I had TB, but by then, I had lost everything. All of this happened because I wasn't aware of the symptoms of TB.*" (Male, 55 years old, Mbeya Region).

> "*I didn't know what type of disease I had. I ended up selling everything I owned, including a mattress, a radio, and a plot of land, just to access the treatment I sought from various traditional healers, but to no avail.*" (Male, 55 years old, Mbarali District, Mbeya Region).

Furthermore, the COVID-19 pandemic contributed to delays in TB diagnosis and treatment, as overlapping symptoms between TB and COVID-19 created confusion among patients. These symptoms overlap, combined with widespread fear of quarantine or isolation, caused many individuals to avoid seeking care at HCFs. Although the majority of participants were enrolled prior to the pandemic, a few were enrolled in March 2020 and later participated in the qualitative component of the study. These participants reported that during the COVID 19 outbreak, fear of being misidentified as having COVID-19 discouraged people from visiting HCFs when experiencing TB-related symptoms. This reluctance contributed to delays in diagnosis and treatment as well as continued transmission of TB within a community. The following quote illustrates this:

> "You know what, People were scared to go to the hospital when they had a cough. Everyone thought it was corona, and they didn't want to be taken for quarantine, so they stayed home and kept spreading the disease". (Male, 37 years, Songwe Region).

**Coping strategies and financial implications for managing PTB symptoms**

Several respondents reported resorting to the disposal of personal or family/household valuable assets to raise funds needed to settle service fees charged by HCPs. This coping strategy was particularly common among patients seeking care from private HCFs, where user fees were perceived as excessively high and often unaffordable for the less privileged individuals to afford, resulting in substantial OOP payments. Similarly, some traditional healers were reported to set user-fees beyond the means of underprivileged clients. As a result, these clients and their families frequently resorted to paying in-kind, offering valuable property such as tamed animals, poultry, or portions of farm crops believed to hold monetary value equivalent to the required fees. This often necessitated finding immediate buyers for these assets. When buyers were unavailable, patients or their relatives sometimes transferred ownership of such property directly to the service provider through mutual agreement. In these cases, the assets were often sold at significantly reduced prices, causing psychological distress and financial loss for the disposers.

## Rural-urban differences in TB care accessibility

Some patients also delayed attending formal HCFs due to living in remote areas with limited or no access to healthcare services. This was particularly common among those in rural settings, where such services are often limited or unavailable. As a result, these patients frequently postponed seeking care at formal HCFs for PTB diagnosis and timely treatment initiation. The following quote illustrates this experience;

> "*I lived in a remote rural area where there was no nearby health facility. Because of the long distance and lack of accessibility healthcare services in our village, it took me almost a whole year of suffering with this illness before I finally decided to go to the facility and was diagnosed with tuberculosis.*" (Male, 43 years, Songwe Region).

## Perceived quality of care related costs

Respondents expressed dissatisfaction with certain HCFs they visited, citing an inability to access the expected standard of care. Upon further probing, participants explained that many of these facilities lacked reliable laboratory services necessary for the diagnosis of PTB, such as sputum or chest X-ray imaging. Although some HCFs had designated PTB clinics or units, they were often under resourced lacking either necessary diagnostic equipment or skilled personnel to perform accurate testing. These deficiencies often compelled patients to resort to self-medication, using pharmaceuticals obtained from local retail outlets, traditional herbal remedies, or both. Consequently, the delays in accessing the appropriate and timely care exacerbated the progression of PTB, ultimately worsening patient's health conditions and leading to increased OOP expenditures.

> "*Some facilities say they treat TB, but when you go there, they don't have the lab or staff to test TB, no X-ray. So, people end up treating themselves with herbs or buying medicine from shops. This contributes to delays real treatment, makes you sicker, and costs even more in the end*" (Female, 40 years, Songwe Region

## Discussion

This study is among the few that employ a mixed methods research design to explore the costs incurred by individual diagnosed with PTB before the initiation of formal TB treatment in Tanzania. The findings reveal that TB patients and their families incur substantial monetary and non-monetary burdens before receiving a confirmed diagnosis and beginning treatment. These costs both direct and indirect can become catastrophic over the course of patient's journey [13].

A key contribution of these costs is the lack of awareness regarding TB symptoms, which often results in delayed healthcare seeking behaviour. Many patients do not initially recognize the need for formal medical evaluation, opting instead for informal treatment methods without a confirmed diagnosis [39]. Participants also reported that many HCFs with diagnostic and trained staff are located far from their communities, necessitating lengthy and expensive travel that adds to the overall financial burden. As a results individual often resort to self-medication or consult informal healthcare providers, both of which can lead to delayed diagnosis and higher cumulative costs [40]. Time based costs which, despite their economic value, are often unquantified and undocumented, along with in-kind payments made to traditional practitioners or expenses related to self-medication with unregulated drugs, remain a persistent challenge in the context of TB care in LMICs. These challenges potentially contribute to life-threatening or permanent health complications, such as lung damage or disability [41,42]. WHO documentation underscores that early detection of TB and the initiation of correct treatment regime are critical to prevent poor clinical outcomes and to reduce transmission [43,44].

The delays in seeking care and the tendency to consult multiple healthcare providers before receiving a formal TB diagnosis as noted from the data obtained under the current study confirm the already identified inefficiencies in the

current TB detection and notification system in Tanzania [7]. Participants in this study also expressed dissatisfaction with the quality of care at certain HCFs, especially those lacking essential diagnostic tools such as sputum microscopy and chest X-ray imaging [45]. This perceived inadequacy not only contributed to diagnosis delays but also prompted some of the patients to pursue alternative forms of treatment, further complicating their clinical condition and increasing their financial burden [46].

The average pre-treatment total cost per TB patient comprises direct medical, direct non-medical, and indirect costs. These costs varied depending on the number and type of healthcare providers visited [47]. As patients sought care from multiple providers, their total expenditures increased, indicating that provider utilization plays a significant role in the overall economic burden of TB care. This emphasizes the need for more efficient healthcare coordination to minimize unnecessary expenses.

In this study, the majority of individuals diagnosed with TB were unemployed or low-income earners [33,37]. The average pre-treatment cost (USD 28.80) was equivalent to approximately 15.5% of the monthly national minimum wage (USD 186). However, among those who were unemployed and without a stable income at the time of treatment initiation (158 participants; 63%) the economic impact of pre-treatment health seeking was likely far more severe. These findings underscore the significant financial hardship associated with accessing TB care in low-income settings and highlight the urgent need for cost-reducing interventions and health policies that protect economically vulnerable populations [34]

A monthly income framework was applied in this analysis, as it is more accurately reflects the immediate financial burden experienced during the pre-treatment period. In contrast, using annual income figures may underestimate the short-term economic impact of healthcare expenses, particularly in the pre-treatment phase when individuals are often seeking multiple providers and have not yet accessed formal care. In this study, the pre-treatment cost of TB care was USD 28.80. For individual with a monthly minimum national wage of USD 186, this represents approximately 15.5% of their monthly income. Although does not exceed the WHO threshold for catastrophic health expenditure, defined as healthcare costs exceeding 20% of annual household income or expenditure, it still constitutes a significant financial burden [48]. For individual without employment or regular income, this cost is likely catastrophic [33]. Therefore, despite falling below WHO's annual threshold, the 15.5% burden at the monthly level highlights a critical gap in financial risk protection for low-income and vulnerable populations. These findings highlight a major access barrier in the early stage of care seeking, where financial constraints can delay or prevent timely diagnosis and treatment. Such delays may worsen disease progression, increase morbidity, and contribute to continued transmission within the community [33].

In addition, the COVID-19 pandemic introduced multifaceted challenges to health systems worldwide, particularly in the diagnosis and management of chronic infectious diseases such as TB. Our findings reveal that the overlapping clinical features of TB and COVID-19 significantly contributed to delays in seeking medical care and treatment initiation for TB. This is consistent with studies conducted in high-burden TB settings, which have reported diagnostic ambiguity due to shared symptoms such as cough, fever, and respiratory distress, resulting in hesitancy among patients to seek formal healthcare services [49,50]. The intersection of TB and COVID-19 has thus exacerbated pre-existing health disparities. Patients delayed care due to fear of isolation, stigma, and the direct economic costs associated with testing and treatment. These factors not only led to disease progression in individual cases but also likely increased community-level transmission due to the extended infectious period in undiagnosed individuals. Moreover, this delay in treatment initiation further strained already burdened TB control programs, complicating efforts to meet global targets for TB elimination [51].

Improving the quality services in Tanzania is vital for enhancing patient outcomes and reducing unnecessary financial and time related burden. While expanding the availability and geographic reach of health facilities is important, it is not sufficient unless these facilities are also adequately staffed with trained personnel and equipped with essential diagnostic tools and supplies. Strengthening quality assurance mechanisms is therefore critical to ensure adherence to standard clinical practices. In parallel, ongoing training and capacity building for healthcare providers especially in rural and underserved areas can support early detection of TB, reduce diagnostic delays, and improve treatment outcomes [46].

To address these challenges more effectively, national TB programs should adopt a multi-pronged approach. First, increasing public health education and awareness is essential, particularly as not all frontline HCWs are sufficiently informed. This effort should extend to informal drug retailers, who may lack knowledge of early TB's symptoms and contribute to inappropriate self-medication without a confirmed diagnosis [27]. Social protection measures such as subsidies for TB diagnosis services (e.g., chest X-rays), transportation allowances, or direct financial assistance could alleviate the economic burden on TB patients, promote earlier diagnosis and improve health outcomes [52]. Furthermore, the overlap in respiratory symptoms between TB and COVID-19 created significant barriers to care seeking during pandemic, not simply due to misclassification, but more importantly due to patients' fear of being diagnosed with COVID-19 and subjected to isolation, stigma, and additional burdens. This highlights the need for integrated screening approaches where individuals presenting with respiratory symptoms are tested for both TB and COVID-19, as well as for public health emergency responses that are sensitive to these dynamics. Policy makers, healthcare providers, and the public must be equipped with accurate information and stigma prevention strategies to avoid unintended consequences for TB care during future outbreaks.

Given the prominent role of traditional medicine in many Tanzanian communities [53], integrating traditional healers into the formal healthcare system present a valuable opportunity. Training these practitioners to recognize TB symptoms and establishing referral pathways to formal HCFs could improve access to care and reduce diagnostic delays. In this role, traditional healers can act as community-level health allies, helping to identify suspected TB cases and ensure timely referrals. Raising community awareness about the importance of early diagnosis and treatment may also reduce reliance on self-medication and informal care-seeking practices [27,54]. Community-based outreach, targeted educational campaigns, and increased engagement of CHWs are remain promising strategies to address these persistent challenges.

## Strengths and limitations

A major strength of this study lies in its use of a mixed-methods research design. The integration of quantitative data, obtained through structured questionnaire, with qualitative insights from open-ended questions allowed for a richer interpretation of the findings. This triangulation enhanced the contextual understanding of the statistical results, enabling us to support key arguments with first hand perspectives from respondents.

The study also provides relevant insights for the national TB program in Tanzania by shedding light on the TB pretreatment costs-related barriers that hinder timely initiation of treatment among eligible patients in the study regions. These findings align with reports from other parts of the country [55,56]. Although TB diagnosis and treatment services are officially provided free of charge under national policy, this study highlights those direct costs particularly those associated with transportation to and from HCFs pose a substantial burden [57]. These barriers suggest a need to decentralize TB services and bring them closer to communities in need. Without such measures, many individuals may remain excluded from the benefits of so-called 'free' services. The analysis of cost related delays in the TB care pathways provides valuable information for refining TB policy and program implementation both in Tanzania and similar settings [58].

Despite its valuable contributions, this study has some limitations. Although social science techniques were employed to encourage openness and accuracy in responses, recall bias may have influenced the reliability of the data, as participants were asked to reflect on experiences that occurred over extended periods. To mitigate recall bias, structured techniques such as time anchoring and sequencing were applied to help participants accurately reconstruct their care-seeking pathways from symptom onset to diagnosis. Moreover, data from the TB Sequel cohort indicate that the median time between symptom onset and the baseline interview was approximately 8 weeks (IQR: 4–16 weeks), representing a relatively short recall period. This strengthens confidence in the reliability of participants' reports of out-of-pocket expenditures and time-related costs.

**Selection bias is another** limitation of this study, stemming from the recruitment of participants through the ongoing TB Sequel project. This approach may limit the generalizability of the findings, as it primarily captures the experiences of

individuals who are enrolled in and retained within structured research and care programs. Consequently, it may exclude people with TB who were never enrolled, were lost to follow-up, or are not currently engaged in care groups whose experiences and cost burdens may differ substantially. To mitigate this limitation, purposive sampling was employed to ensure diversity within the participant pool, taking into account geographic variation (across districts in both study regions), urban and rural settings, gender, and the inclusion of individuals living with TB/HIV co-infection. Additionally, several steps were taken to minimize the risk of recall bias and its impact on the accuracy of reported costs and timelines.

Income and Employment Instability: A significant portion of the study population was unemployed or earning irregular income, which posed challenges in accurately assessing economic burden. While we used the national minimum wage as a proxy for income, this may not fully capture the diverse financial realities of participants, potentially leading to under- or overestimation of relative costs.

The qualitative data collection for this study was notably impacted by the COVID-19 pandemic (2020–2021), as lockdowns, travel restrictions, and the strain on healthcare systems hindered timely and comprehensive fieldwork. These disruptions limited access to participants particularly in remote or underserved areas and reduced opportunities for in-person interviews, which are essential for capturing rich, context-specific insights. In addition, the pandemic led to a nationwide decline in TB case reporting, affecting both the availability and representativeness of potential participants. Many individuals who might have been included in the study were likely undiagnosed or avoided healthcare services due to fears of COVID-19 exposure or quarantine, introducing potential selection bias and limiting the generalizability of the findings. Despite these constraints, the study provides valuable perspectives on the intersection of TB care and public health emergencies, highlighting the importance of flexible and resilient data collection approaches during periods of systemic disruption.

Moreover, the study focused primarily on patients with PTB, and did not capture costs associated with, extra-pulmonary TB nor on HIV-AIDS or other lung and non-lung related diseases normally faced by the patients, including those experiencing comorbidities [59] . These comorbidities could contribute to higher healthcare costs, which were not fully captured in our analysis.

## Conclusion

This study reaffirms the significant challenges faced by individuals and households in Tanzania, when seeking healthcare for TB related illness before formal diagnosis. Both financial burdens such as transportation and opportunity costs, and non-financial burdens, persist across formal and informal care-seeking pathways. These delays contribute to disease progression, ongoing transmission, and significant long-term financial hardship for the affected families. Although the National TB program promote early detection and free TB services under Universal Health Coverage, the findings underscore the need for stronger integration with traditional healthcare practices. Introducing targeted pre-diagnosis support mechanisms such as fee waivers or transport subsidies, could help improve access to timely care. Addressing these barriers through a comprehensive, multisectoral approach is critical to improve TB treatment outcome and advancing public health equity at both the individual and community levels.

## Supporting information

**S1 Checklist.  Inclusivity in global research.**
(DOCX)

## Acknowledgments

We would like to appreciate the support of the TB Sequel Project staff, specifically Dr. Julieth Lalashowi and Dr. Elimina Siyame.

## Author contributions

**Conceptualization:** Stella Kilima, Godfrey Mubyazi, Denise Evans.

**Data curation:** Stella Kilima, Kamban Hirasen, Godfrey Mubyazi, Nyanda Ntinginya, Issa Sabi, Simeon Mwanyonga.

**Formal analysis:** Stella Kilima, Kamban Hirasen, Godfrey Mubyazi, Aneesa Moolla.

**Funding acquisition:** Nyanda Ntinginya, Issa Sabi, Denise Evans.

**Investigation:** Stella Kilima, Simeon Mwanyonga.

**Methodology:** Kamban Hirasen, Godfrey Mubyazi, Aneesa Moolla, Denise Evans.

**Project administration:** Stella Kilima, Nyanda Ntinginya, Issa Sabi, Simeon Mwanyonga.

**Software:** Stella Kilima, Aneesa Moolla, Nyanda Ntinginya, Issa Sabi, Denise Evans.

**Supervision:** Godfrey Mubyazi, Aneesa Moolla, Denise Evans.

**Validation:** Stella Kilima, Kamban Hirasen, Godfrey Mubyazi, Aneesa Moolla, Issa Sabi, Simeon Mwanyonga, Denise Evans.

**Writing – original draft:** Stella Kilima, Kamban Hirasen, Godfrey Mubyazi, Aneesa Moolla, Nyanda Ntinginya, Denise Evans.

**Writing – review & editing:** Stella Kilima, Kamban Hirasen, Godfrey Mubyazi, Aneesa Moolla, Nyanda Ntinginya, Issa Sabi, Simeon Mwanyonga, Denise Evans.

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
