## [Decision Letter · Decision Letter 0]

15 Nov 2024

Thank you for submitting your manuscript to PLOS ONE. After careful consideration, we feel that it has merit but does not fully meet PLOS ONE’s publication criteria as it currently stands. Therefore, we invite you to submit a revised version of the manuscript that addresses the points raised during the review process.

We look forward to receiving your revised manuscript.

Kind regards,

Bilal Ahmad Rahimi, M.D., D.T.M.&H., M.C.T.P., Ph.D

Academic Editor

PLOS ONE

Journal Requirements:

2. Please include a complete copy of PLOS’ questionnaire on inclusivity in global research in your revised manuscript. Our policy for research in this area aims to improve transparency in the reporting of research performed outside of researchers’ own country or community. The policy applies to researchers who have travelled to a different country to conduct research, research with Indigenous populations or their lands, and research on cultural artefacts. The questionnaire can also be requested at the journal’s discretion for any other submissions, even if these conditions are not met. Please find more information on the policy and a link to download a blank copy of the questionnaire here: https://journals.plos.org/plosone/s/best-practices-in-research-reporting . Please upload a completed version of your questionnaire as Supporting Information when you resubmit your manuscript.

3. Thank you for stating the following financial disclosure: [This work was supported by the German Ministry for Education and Research (GMBF) through the TB Sequel project under grant [LMU-IMPH-TB Sequel-01]].

4. We note that you have indicated that there are restrictions to data sharing for this study. For studies involving human research participant data or other sensitive data, we encourage authors to share de-identified or anonymized data. However, when data cannot be publicly shared for ethical reasons, we allow authors to make their data sets available upon request. For information on unacceptable data access restrictions, please see http://journals.plos.org/plosone/s/data-availability#loc-unacceptable-data-access-restrictions .

b) If there are no restrictions, please upload the minimal anonymized data set necessary to replicate your study findings to a stable, public repository and provide us with the relevant URLs, DOIs, or accession numbers. Please see http://www.bmj.com/content/340/bmj.c181.long for guidelines on how to de-identify and prepare clinical data for publication. For a list of recommended repositories, please see https://journals.plos.org/plosone/s/recommended-repositories . You also have the option of uploading the data as Supporting Information files, but we would recommend depositing data directly to a data repository if possible.

6. We note that Figures 1 and 2 in your submission contain [map/satellite] images which may be copyrighted. All PLOS content is published under the Creative Commons Attribution License (CC BY 4.0), which means that the manuscript, images, and Supporting Information files will be freely available online, and any third party is permitted to access, download, copy, distribute, and use these materials in any way, even commercially, with proper attribution. For these reasons, we cannot publish previously copyrighted maps or satellite images created using proprietary data, such as Google software (Google Maps, Street View, and Earth). For more information, see our copyright guidelines: http://journals.plos.org/plosone/s/licenses-and-copyright .

1. You may seek permission from the original copyright holder of Figures 1 and 2 to publish the content specifically under the CC BY 4.0 license.

We recommend that you contact the original copyright holder with the Content Permission Form (http://journals.plos.org/plosone/s/file?id=7c09/content-permission-form.pdf ) and the following text:

“I request permission for the open-access journal PLOS ONE to publish XXX under the Creative Commons Attribution License (CCAL) CC BY 4.0 (http://creativecommons.org/licenses/by/4.0/ ). Please be aware that this license allows unrestricted use and distribution, even commercially, by third parties. Please reply and provide explicit written permission to publish XXX under a CC BY license and complete the attached form.”

Reviewers' comments:

Reviewer's Responses to Questions

**Comments to the Author**

1. Is the manuscript technically sound, and do the data support the conclusions?

Reviewer #1: Yes

Reviewer #2: Partly

2. Has the statistical analysis been performed appropriately and rigorously?

Reviewer #1: Yes

Reviewer #2: No

3. Have the authors made all data underlying the findings in their manuscript fully available?

Reviewer #1: No

Reviewer #2: Yes

4. Is the manuscript presented in an intelligible fashion and written in standard English?

Reviewer #1: Yes

Reviewer #2: Yes

Reviewer #1: The manuscript would benefit from additional editing to harmonise terminology and be more concise and to the point. State "mixed methods study" clearly in Methodology

References to global and national TB burden source from various Global TB reports between 2019-2023. Please avoid this pick-and-choose and stick to the latest data.

Several aspects missing:

- The COVID-19 impact on the situation/study

- Definition of catastrophic cost and current estimated prevalence (WHO: 49%) - central to this manuscript

- Context of costs incurred - value against benchmark (average weekly, monthly or annual income.)

Some statements are incorrect, open to misinterpretation, or miss-quoted:

- Tanzania,[...] has high cases of TB in comparison to other countries within Africa and globally - better stick to is one of the 30 high TB burden countries, as there are other countries with much higher TB burden in Africa & globally

- A recent national TB prevalence survey in Tanzania revealed that 145 out of every 100,000 tuberculosis cases were detected - This is a miss-quote – the case detection rate is 145/100,000 population not 145 out of 100,000 tuberculosis patients

- Four patients initiated a MDR regime because they were Multidrug-resistant (MDR). - People are not MDR-TB, they have MDR-TB. Please ensure non-stigmatising respectful language

Methodology:

- The criteria for the sampling strategy are missing, as are sampling bias mitigation. Missing insight into how many people have declined participation.

- Why were the questionnaires orally translated rather than fully translated in writing (ensuring consistency of translation)?

Table 5. Themes & quotes from participants: It would be better to add the quotes to the respective thematic paragraphs to underline the stated “findings”/ interpretations

At present the format is a bit disjointed and previously stated “findings” are missing the actual data/ information/ quotes underpinning them

Discussion: reference some more countries

Reviewer #2: Implications of pre-diagnosis costs incurred by patients and their families for tuberculosis-related health-seeking behaviours in Mbeya and Songwe regions, Tanzania

General comment

The paper analyses costs associated with pre-diagnosis treatment in Mbeya and Songwe regions of Tanzania. Implications of pre-diagnosis costs incurred by patients and their families for tuberculosis-related health-seeking behaviours in Mbeya and Songwe regions, Tanzania. The study uses data collected from a nested cohort of 261 PTB patients over a period of time, to gather patient out-of-pocket (OOP) costs, in order to analyse the implications for health seeking behaviour.

The study presents new data on OOP faced by patients diagnosed with PTB in a LMIC setting. The study design was well designed, and the data were collected using well designed instruments. It appears that that surveys were well executed. The data are well analysed. This is a new empirical study on a topic of great academic and policy interest, especially in the context of UHC.

The paper is generally well written. However, I have some comments that i would request the authors to think about as they revise their manuscript.

Background

The reference about OOPs are all related to diagnosed TB cases rather than pre-diagnosed cases. Is there no data on costs associated with pre-diagnosis PTB?

Please find some references for this statement: “While many high TB burden countries offer essential TB diagnosis and treatment for free...

The motivation for the paper could be strengthened. In the second paragraph of the Background, the authors state that the economic burden of PTB is significant, but provide no reference or quantification of their claim. They provide a reference based on Thailand only for OOPs, while there are various better sources to back up this statement of the economic burden of PTB, that could include Africa as well. I would urge the authors to strengthen this point, as it is crucial to motivating their paper.

I think that the background needs to be recast to reflect the focus of the paper on economic costs associated with pre-diagnosis PTB. In its current form, it is a mix of costs experienced before, during and post diagnosis. Is there an argument for pre-diagnosis costs? If so, a specific motivation for pre-diagnosis economic costs should be presented in the background.

The study needs to do better job at describing why PTB pre-treatment costs are important. What do we know about them, why do we need to care?

TB services in Tanzania

The authors say that “prevalence survey in Tanzania revealed that 145 out of every 100,000 tuberculosis cases were detected”. Is this per TB cases or per population? And how is progress being assessed with one data point? I am sure that authors know something about the trajectory of TB burden in Tanzania. Let them clarify this point.

I would also urge the authors to provide further details about where TB services are delivered in Tanzania. They have said that it is in the public sector. But, no details about the level of care, referral system, level of access (what percentage of patients receive care in the public system, etc.), quality, etc.

Do all the health facilities have capacity to diagnose PTB? In the analysis, it is shown that patients incurred significant expenses while visiting various providers, partly because they were unaware which provider had the capacity to diagnose them.

Methods

The patients were followed up over a relatively long period of time. A key concern I have is that given that data collected retrospectively over time, what is the potential effect of recall bias?

What was the median period in months or weeks, between onset of symptoms to initiation of treatment? How long was the diagnosis?

What were the inclusion and exclusion criteria for the selection of subjects, apart from age and TB test?

From which health facilities were these patients being referred from?

Findings

Overall, the descriptive analysis is well presented. I just have a few comments in this section.

• It would be interesting, if possible, for the authors to run a multivariate analysis of costs per patient characteristics. Does sex, HIV status, residential location, lengthy of delay in seeking treatment, etc., have anything to do with the level of reported costs? The authors have said that the reasons they did not perform such analysis was “to avoid overloading the current paper with excessive data and discussions, we decided to omit some detailed quantitative data analysis”. I am not sure that this a convincing reason.

• Could the authors comments on what are these drugs that patients taking pre-diagnosis?

• Given that the data were gathered over a period of time, it would have been helpful to see the economic costs calculated over a specified period of time, say per month or six months, rather than aggregated over the entire sample. That would also provide a quality check of the potential recall bias that I talked about earlier.

• I would suggest to strengthen the analysis by putting these costs into some perspective for ease of interpretation. For example, how do they relate to per capita income, etc.

Discussion

This section needs some attention to provide a clear storyline about economic costs of PTB pre-diagnosis. In its current form, the discussion does not communicate very well the interpretation of the study findings. As an example, I am not cure what is meant by the following statements: “Therefore, regrets arise after individuals or their households/families opt to use medicines that are obtained outside of formal prescriptions by qualified medical officers, resulting in certain monetary and non-monetary costs.”, and also “Opting for shortcuts in TB care-seeking behaviours is a challenge that is faced in both studied regions. Initially, the respective parties believe that by opting for shortcuts in TB care seeking, they can continue with their daily activities, such as participating in economically productive work as farmers or other jobs that provide a means of subsistence”. What exactly are these shortcuts?

The authors could also provide further discussion of the policy implications of these findings. They should attempt to link policy implications to the specific findings of their study. In several parts, this discussion is sounding more speculative than rooted in the findings of the study. As an example, I did not see where this recommendation came from in the analysis: “Additionally, national TB control programs should make necessary adjustments, such as designing and institutionalising social protection options to support the less privileged and marginalised individuals and their families”.

What are the implications for strengthening key aspects of the TB in Tanzania: notification, screening, detection, treatment, etc. This paper is about implications, perhaps more attention could be given to the health system side of things.

Conclusion

I think that this section needs to be re-written and shortened. The focus should be the evidence found in terms of the answer to the question posed or objective. What is the scale or scope of economic costs of PTB in Tanzania? Some of the points in the second and third paragraphs could go to the discussion section.

Minor comments

This sentence, However, its burden is higher in specific countries, Tanzania being a no-exclusion., could be simplified to read: However, its burden is higher in specific countries, Tanzania being one such country.

Writing style needs some attention. For example, use more conventional language for this statement: “As mentioned above, the public health implications of TB are dire when examined both broadly and in depth.” Terms such as dire, immense, etc., are not conventional scholarly words in scientific writing.

Would the authors want to clarify or simply this sentence: “Therefore, in the absence of some safety net mechanisms, such as user fee exemptions or health insurance, patients and their families find themselves in access to care financial hardships, contrary to the UHC goal and the global End-TB Strategy”

**Do you want your identity to be public for this peer review?** For information about this choice, including consent withdrawal, please see our Privacy Policy

Reviewer #1: **Yes: ** Kathy Fiekert, MSc, DCHD, DGN, Team Lead Health & Community Systems, KNCV TB Plus

Reviewer #2: No

---

## [Author Response · Author response to Decision Letter 1]

10 Jul 2025

General Comment: We note that Figures 1 and 2 in your submission contain [map/satellite] images which may be copyrighted. All PLOS content is published under the Creative Commons Attribution License (CC BY 4.0), which means that the manuscript, images, and Supporting Information files will be freely available online, and any third party is permitted to access, download, copy, distribute, and use these materials in any way, even commercially, with proper attribution. For these reasons, we cannot publish previously copyrighted maps or satellite images created using proprietary data, such as Google software (Google Maps, Street View, and Earth). We require you to either (1) present written permission from the copyright holder to publish these figures specifically under the CC BY 4.0 license, or (2) remove the figures from your submission.

Response: Thank you for your guidance regarding the copyright requirements for Figures 1 and 2 in our submission. In compliance with PLOS ONE's licensing policy and the CC BY 4.0 requirements, we have removed Figures 1 and 2 from the manuscript. We understand the importance of ensuring that all published content is fully compatible with the journal’s open-access licensing terms. We have updated the manuscript and figure number accordingly.

Reviewer 1

1. The manuscript would benefit from additional editing to harmonize terminology and be more concise and to the point.

Response: Thank you for your valuable feedback. We appreciated your suggestion for additional editing to harmonize terminology and improve conciseness. We carefully reviewed the manuscript to ensure consistent terminology throughout and revised the text to make it more concise and to the point, while maintaining clarity and accuracy in the content. Your input helped strengthen the overall quality of the manuscript (whole document).

2. State "mixed methods study" clearly in Methodology

References to global and national TB burden source from various Global TB reports between 2019-2023. Please avoid this pick-and-choose and stick to the latest data.

Response: Thank you for your constructive feedback. In response, we clearly stated that the study employed a mixed methods approach in the Methodology section to enhance clarity.

We have revised the Methodology section to clearly state the mixed methods study. The revised sentence now reads: “This study employed a mixed methods approach, incorporating both quantitative and qualitative components

Additionally, we ensured that all references to the global and national TB burden were based on the latest available data, drawn from Global TB reports between 2019 and 2023, to avoid inconsistencies. We appreciated your suggestion and made the necessary adjustments to improve the manuscript (Page 2 and 7. Lines 30; 141-142).

3. Several aspects missing: - The COVID-19 impact on the situation/study

Response: Thank you for your valuable feedback. In response, we included a discussion on the impact of COVID-19 on the study, acknowledging its potential effects on the results and the broader context. We addressed how the pandemic may have influenced the TB burden and any changes in data patterns, ensuring that this aspect is thoroughly covered in the revised manuscript (Result section Page 24- Lines 472-481) (Discussion section page 28- Lines 569-576) (Study limitations page 31- Lines 635-644).

4. Definition of catastrophic cost and current estimated prevalence (WHO: 49%)

Response: Thank you for providing a more detailed explanation. We revised the manuscript to incorporate your precise definition of catastrophic costs. Here's a refined version of the definition:"Catastrophic costs occur when total costs of TB care seeking including direct medical, direct non medical and indirect costs exceed 20% of household’s annual income prior to TB diagnosis.This definition and prevalence are clearly articulated in the manuscript, with appropriate citation of the WHO reference. Thank you again for highlighting this point. (Page 5, Background section - Lines 92-94)

5. central to this manuscript - Context of costs incurred - value against benchmark (average weekly, monthly or annual income.) Some statements are incorrect, open to misinterpretation, or miss-quoted:- Tanzania,[...] has high cases of TB in comparison to other countries within Africa and globally - better stick to is one of the 30 high TB burden countries, as there are other countries with much higher TB burden in Africa & globally,

Response: We appreciate the reviewer’s emphasis on contextualizing the costs incurred by TB patients. In response, we have updated the literature review section to include comparative data from other African countries, thereby highlighting the substantial share of pre-diagnosis costs in relation to household income and time loss. (Page 4 &5; Literature review section - Lines 75-83). Additionally, in the discussion section, we now provide a clearer comparison of pre-treatment costs against the national minimum monthly wage to better illustrate the financial burden, particularly for low-income populations (Lines 558-568). Regarding the statement on TB burden, we appreciate your clarification. We have revised the wording to align with WHO classifications and now state “Tanzania is one of the 30 high TB burden countries globally” (Background section 4- Lines 100-101).

6. A recent national TB prevalence survey in Tanzania revealed that 145 out of every 100,000 tuberculosis cases were detected - This is a miss-quote – the case detection rate is 145/100,000 population not 145 out of 100,000 tuberculosis patients

Response: Thank you for pointing out the misquote. We apologize for the error in phrasing. We have revised the statement to correctly reflect that the case detection rate for tuberculosis in Tanzania is 145 per 100,000 population, not per 100,000 tuberculosis patients. We appreciate your careful attention. (Page 6 Background section- Lines 104-106)

7. Four patients initiated a MDR regime because they were Multidrug-resistant (MDR). - People are not MDR-TB, they have MDR-TB. Please ensure non-stigmatising respectful language.

Response: Thank you for the important feedback. We fully agree with the need for respectful and non-stigmatizing language. The sentence has been revised to: "Four patients initiated a MDR-TB regimen because they had multidrug-resistant tuberculosis (MDR-TB)." We appreciate your guidance on this and have made the necessary adjustments to ensure respectful language throughout the manuscript (Page 15 Result section - Lines 313-315)

8. Methodology:

The criteria for the sampling strategy are missing, as are sampling bias mitigation.

Response: Thank you for your valuable comment. We apologize for the oversight. We have now added a detailed explanation of the criteria used for our sampling strategy in the revised manuscript. Specifically, we clarify the following aspects of our sampling approach:

The study employed random sampling, to ensure representative selection. Additionally, we have addressed potential sampling biases and have outlined the strategies employed to minimize them. For qualitative sampling, purposively selection were used. (Page 8 Methodology section - Lines 160-165 and 184-186).

9. Missing insight into how many people have declined participation.

Response: Thank you for your comment. A total 319 participants were initially recruited. Of these, 49 did not meet the eligibility criteria, 6 tested negative for TB, and 3 declined to proceed with follow-up procedures. Consequently, 261 participants with laboratory-confirmed PTB patients, representing 81% of those initially recruited, were enrolled in the TB Sequel study (Page 15 result section - Lines 301-305)

10. Why were the questionnaires orally translated rather than fully translated in writing (ensuring consistency of translation)?

Response: Thank you for your observation. We acknowledge the distinction in translation approaches between the quantitative and qualitative tools. The quantitative questionnaire was administered through oral translation by trained bilingual researchers to accommodate real-time interview conditions across diverse settings (Quantitative Page 10 methodology section - Lines 197-200).

In contrast, the qualitative interview guide was fully translated into written Kiswahili to ensure consistency in probing and depth of discussion across all interviews. This written translation was independently verified by two bilingual researchers for accuracy and reliability.

(Qualitative Page 11 methodology section- Lines 221-223)

11. Table 5. Themes & quotes from participants: It would be better to add the quotes to the respective thematic paragraphs to underLines the stated “findings”/ interpretations. At present the format is a bit disjointed and previously stated “findings” are missing the actual data/ information/ quotes underpinning them

Discussion: reference some more countries.

Response: Thank you for this helpful suggestion. We agree that integrating the quotes directly into the thematic paragraphs enhances the clarity and supports the findings more effectively. In response to your feedback, we have revised the manuscript by embedding the relevant quotes within the corresponding thematic sections, ensuring that the quotes directly align with the findings and interpretations. This change aims to create a more cohesive structure and provide a clearer link between the qualitative data and the themes discussed. We hope this improves the readability and coherence of the analysis. (Results section qualitative- Lines 376-526)

Reviewer 2

1. The reference about OOPs are all related to diagnosed TB cases rather than pre-diagnosed cases. Is there no data on costs associated with pre-diagnosis PTB?

Response: Thank you for your comment and observation. You’re correct that many references on OOP costs tend to focus on diagnosed TB cases. However, in several studies, pre-treatment and on-treatment costs are often combined and reported together as part of the overall financial burden, particularly when calculating catastrophic costs. This makes it challenging to find studies that isolate costs strictly associated with the pre-diagnosis phase. (The whole background section)

2. Please find some references for this statement: “While many high TB burden countries offer essential TB diagnosis and treatment for free...

Response: Thank you for your valuable feedback. I appreciate your suggestion to provide references for the statement: However, the statement has been revised to read “Although TB diagnosis and treatment are provided free of charge in many public health systems, patients often incur substantial out-of-pocket expenses, especially during the pre-diagnosis phase of care” The following references was used to inform this statement.

Assefa DG, Dememew ZG, Zeleke ED, Manyazewal T, Bedru A. Financial burden of tuberculosis diagnosis and treatment for patients in Ethiopia: a systematic review and meta-analysis. BMC Public Health. 2024;24(1):1–14.

WHO. Global tuberculosis report 2023. Geneva: World Health Organization; 2023. Licence: CC BY-NC-SA 3.0 IGO; 2023 (Page 4, Background section- Lines 66-68)

3. The motivation for the paper could be strengthened. In the second paragraph of the Background, the authors state that the economic burden of PTB is significant, but provide no reference or quantification of their claim. They provide a reference based on Thailand only for OOPs, while there are various better sources to back up this statement of the economic burden of PTB, that could include Africa as well. I would urge the authors to strengthen this point, as it is crucial to motivating their paper.

Response: Thank you for your valuable feedback. We agree that providing stronger support for the claim regarding the economic burden of pre-diagnosis PTB is crucial for motivating the study. In response to your comment, we have expanded the background section to include more references and data quantifying the economic burden of PTB, particularly with a broader focus on regions such as Africa.

The following references were used to inform: Litvinjenko S, Magwood O, Wu S, Wei X. Burden of tuberculosis among vulnerable populations worldwide: an overview of systematic reviews. Lancet Infect Dis. 2023;23(12):1395–407.

Assefa DG, Dememew ZG, Zeleke ED, Manyazewal T, Bedru A. Financial burden of tuberculosis diagnosis and treatment for patients in Ethiopia: a systematic review and meta-analysis. BMC Public Health. 2024;24(1):1–14.

WHO. Global tuberculosis report 2023. Geneva: World Health Organization; 2023. Licence: CC BY-NC-SA 3.0 IGO; 2023.

Asres A, Jerene D, Deressa W. Pre- And post-diagnosis costs of tuberculosis to patients on Directly Observed Treatment Short course in districts of southwestern Ethiopia: A longitudinal study. J Health Popul Nutr. 2018;37(1):1–11.

Isangula K, Philbert D, Ngari F, Ajeme T, Kimaro G, Yimer G, et al. Implementation of evidence-based multiple focus integrated intensified TB screening to end TB (EXIT-TB) package in East Africa: a qualitative study. BMC Infect Dis. 2023;23(1):1–17.

Jarde A, Romano E, Afaq S, Elsony A, Lin Y, Huque R, et al. Prevalence and risks of tuberculosis income and multimorbidity in low- ¬ income countries : a meta- ¬ review. BMJ Open. 2022;1(12(9):e060906).

Devoid I, Sillah AK, Sutherland J, Owolabi O, Ivanova O, Govathson C, et al. The household economic burden of drug-susceptible TB diagnosis and treatment in The Gambia. International Journal of Tuberculosis and Lung Disease. 2022 Dec 1;26(12):1162–9. (Page 4 Background section - Lines 65-83)

4. I think that the background needs to be recast to reflect the focus of the paper on economic costs associated with pre-diagnosis PTB. In its current form, it is a mix of costs experienced before, during and post diagnosis. Is there an argument for pre-diagnosis costs? If so, a specific motivation for pre-diagnosis economic costs should be presented in the background.

Response: Thank you for your thoughtful comment. We acknowledge that the current background section was not sufficiently focused on the specific topic of pre-diagnosis economic costs associated with pulmonary tuberculosis (PTB), and that it blends costs incurred before, during, and after diagnosis. In light of your feedback, we have revised the background to more clearly emphasize the rationale for focusing on pre-diagnosis costs. (Page 4-7 Background section - Lines 55-138)

5. The study needs to do better job at describing why PTB pre-treatment costs are important. What do we know about them, why do we need to care?

Response: Thank you for your insightful comment. We agree that the importance of pre-treatment costs for PTB needs to be more clearly articulated. In response, we have revised the manuscript to better emphasize the significance of these costs and the reasons they warrant attention. (Page 7, Background section- Lines 128-138)

6. The authors say that “prevalence survey in Tanzania revealed that 145 out of every 100,000 tuberculosis cases were detected”. Is this per TB cases or per population?

Response: Thank you for your valuable comment. We appreciate the opportunity to clarify this point. We have revised the statement to correctly reflect that the case detection rate for tuberculosis in Tanzania is 145 per 100,000 population, not per 100,000 tuberculosis patients. This figure represents the number of people diagnosed with TB per 100,000 members of the general population, as reported in the prevalence survey. (Page 6 Background section - Lines 104-106)

7. And how is progress being assessed with one data point? I am sure that authors know something about the trajectory of TB burden in Tanzania. Let them clarify this point.

Response: Regarding the assessment of progress with a single data point, we recognize that it would be insufficient to gauge the trajectory of TB burden based solely on one prevalence survey. We clarified that the cited data represents the most recent available point in time for the survey, but we also acknowledge that TB burden should be tracked over time to understand progress or setbacks. In the revised manuscript, we have added a brief discussion of th

---

## [Decision Letter · Decision Letter 1]

3 Sep 2025

Dear Dr. STELLA Peter KILIMA,

Thank you for submitting your manuscript to PLOS ONE. After careful consideration, we feel that it has merit but does not fully meet PLOS ONE’s publication criteria as it currently stands. Therefore, we invite you to submit a revised version of the manuscript that addresses the points raised during the review process.

We look forward to receiving your revised manuscript.

Kind regards,

Bilal Ahmad Rahimi

Academic Editor

PLOS ONE

Journal Requirements:

Reviewers' comments:

Reviewer's Responses to Questions

**Comments to the Author**

Reviewer #1: (No Response)

2. Is the manuscript technically sound, and do the data support the conclusions?

Reviewer #1: Partly

3. Has the statistical analysis been performed appropriately and rigorously?

Reviewer #1: Yes

4. Have the authors made all data underlying the findings in their manuscript fully available?

Reviewer #1: No

5. Is the manuscript presented in an intelligible fashion and written in standard English?

Reviewer #1: No

Reviewer #1: My apologies for the late review. Coming late to the process though enabled me to review the revision document. I agree with the comments of my fellow reviewers and am pleased to note that they were well addressed in the revision.

I have some additional comments though:

1. Costs are relative to income, this is why the WHO is defining catastrophic costs as “exceeding 20% of annual household income or expenditure” – therefore the discussion (and the abstract) would benefit from an expression of costs as percentage of the annual income or expenditure. It is also not clear from the manuscript if the USD 28.8 denotes the average total expenditure or the average expenditure per month. Line 41 in the abstract speaks of “average total pre-treatment cost” while line 562 denotes “cost incurred by patients in this study was USD 28.80 per month” Furthermore, there seems to be a typo 28.3 instead of the 28.8 (Table 4 shows a mean of USD 28.8 – so the number in line 41 should be corrected.

Line 41) expression “USD 28.3 (representing x% of average/ minimum annual income)” would be providing helpful context for the reader.

2. Context Line 157) For completeness and transparency, it would be helpful to describe in one sentence what the Sequel study is about. Also to understand that no inadvertent sample bias is included by working with participants selected for another study – e.g. asking participants to travel to another health facility for additional interventions, when they have already incurred increased pre-diagnostic costs and delays – does this influence willingness to participate?

3. Spelling & grammar: The document needs some more work on the copyediting: please carefully check and edit the manuscript as there are still large numbers of spelling and grammar mistakes which might not be picked up by the automated checker (as they are proper words) but distort the meaning of your sentences. Some (not exhaustive) examples are:

Line 28) grammar: suggest removing “were”

Line 45) L missing in “formal”

Line 95) spelling “dater” instead of “deter”

Line 143) please spell out “Sub-Saharan African” on first use

Line 303) please remove “patient” from the end of the line – seems a remnant from editing and is not needed

Line 331)/2) sentence is confusing – why not simply say “have not contacted any other service providers before presenting to the health facility”?

Line 396) Missing “who” after “Among those”

4. Stigmatising language: Line 164) please mitigate stigmatising language HIV-positive => People/ participants living with HIV; diabetic patients => people/ participants with diabetes mellitus.

5. Inclusion criteria: Line 189) it strikes me that the ability to comfortable communicate in English or Kiswahili and provide written consent might exclude people with less or no formal education and thus potentially less socio-economic resources/ security - i.e. at higher risk of being affected by catastrophic costs, thus potentially underestimating the true OOP expenditures. Has this been considered and mitigated? It should at least be acknowledged.

6. Table 1: Line 285/286) the table would benefit from proper lines to better distinguish which sub-themes belong to which theme.

7. P-values Lines 320-322) “Other variables, such as age, sex, education level, HIV status, smear microscopy results, and treatment regimen, show no significant differences in pre-treatment costs, based on the p-values calculated in the tests (Table 2)” - the sentence refers to p-values in table 2 – which do not exist in said table – please rephrase

8. Figure 1: Line 337) Figure 1 would benefit from clearer labelling of the x axis (what do the numbers 1-4 denote?) and some more description. As the values are percentages of visits in each category, but the number of people in each group likely decreases is the figure not slightly misleading? I would at least expect a number of respondents indicated for each group

9. Table 3: Line 349) Table 3 shows that the visit time at provider increases for each subsequent visit, not the travel time - the travel time only increases between 2nd and 3rd visit

10. Discussion: 543/544 I would propose that “worsening condition with potentially worse treatment outcome (e.g. long term-disability) of the affected person and ongoing transmission” should be the first concerns before MDR TB (as long as no TB drugs are use pre-diagnostic, the development of drug resistance is unlikely) – DR TB becomes an issue when Pharmacies/ drug sellers or informal providers provide TB drugs of insufficient strengths and unproven combinations. I would suggest referencing the immediate risks to patients and programme first, as they are touched upon by the respondents already (“I got worse” “I lost all my possessions”) – DR-TB does not seem the main issue in your catchment area, as only 3 out of your 261 participants were diagnosed with/ treated for DR-TB (1%)

11. COVID-19 Line 599-602: by the very nature of things, it is hard to distinguish COVID-19 from TB just by symptomatic assessment – no education programme with change this. The key is that patients expressed fear of being diagnosed with COVID-19 and then having to face isolation and additional stigma. Therefore the 2 logical conclusions would be to a) ensure that anyone presenting with respiratory symptoms is tested for both TB and COVID-19; b) public health emergency measures are designed in a way that they do not frighten people suffering from other diseases away from accessing services. Policy makers, service providers and the public need the right information and stigma prevention measures.

**Do you want your identity to be public for this peer review?** For information about this choice, including consent withdrawal, please see our Privacy Policy

Reviewer #1: **Yes: ** Kathy Fiekert

---

## [Author Response · Author response to Decision Letter 2]

10 Oct 2025

Reviewer 1

1. Costs are relative to income; this is why the WHO is defining catastrophic costs as “exceeding 20% of annual household income or expenditure” – therefore the discussion (and the abstract) would benefit from an expression of costs as percentage of the annual income or expenditure. It is also not clear from the manuscript if the USD 28.8 denotes the average total expenditure or the average expenditure per month. Line 41 in the abstract speaks of “average total pre-treatment cost” while line 562 denotes “cost incurred by patients in this study was USD 28.80 per month” Furthermore, there seems to be a typo 28.3 instead of the 28.8 (Table 4 shows a mean of USD 28.8 – so the number in line 41 should be corrected.

Line 41) expression “USD 28.3 (representing x% of average/ minimum annual income)” would be providing helpful context for the reader

Response: Thank you for this insightful comment. We agree that expressing costs as a percentage of annual income or expenditure can provide valuable context. However, due to the high levels of unemployment observed in our study population (now reflected in Table 2), and the fact that the majority of participants experienced delays in seeking care ranging from 4–8 months with a few delaying up to 16 months, we opted to calculate costs based on the minimum monthly income. This decision was informed by the observation that most respondents reported earning either the national minimum wage or an amount close to it. We chose not to use average annual income as a reference point, as this could be misleading in our context and potentially underestimate the financial burden of pre-treatment costs. For individuals with limited or unstable income, an expenditure of USD 28.80 represents a disproportionately large burden compared to someone with regular and stable earnings.

Page 2 Line 40: Page 21; Line: 377-382; Page 29&30 line 573-597; Page 34 Line 685-689;

2. Comment: For completeness and transparency, it would be helpful to describe in one sentence what the Sequel study is about. Also to understand that no inadvertent sample bias is included by working with participants selected for another study e.g. asking participants to travel to another health facility for additional interventions, when they have already incurred increased pre-diagnostic costs and delays does this influence willingness to participate?

Response: Thank you for this valuable observation. We have now added a brief description of the Sequel study in the Methods section to improve clarity and context. We also acknowledge the potential for selection bias and have clarified in the revised manuscript how participants were recruited.

Page 8; Line 143-145; Page 9: Line 166-178; Page 10 line 193-197; Page 34; line 672-682;

3. Spelling & grammar: The document needs some more work on the copyediting: please carefully check and edit the manuscript as there are still large numbers of spelling and grammar mistakes which might not be picked up by the automated checker (as they are proper words) but distort the meaning of your sentences. Some (not exhaustive) examples are:

Line 28) grammar: suggest removing “were”

Line 45) L missing in “formal”

Line 95) spelling “dater” instead of “deter”

Line 143) please spell out “Sub-Saharan African” on first use

Line 303) please remove “patient” from the end of the line – seems a remnant from editing and is not needed

Line 331)/2) sentence is confusing – why not simply say “have not contacted any other service providers before presenting to the health facility”?

Line 396) Missing “who” after “Among those”

Response: Thank you for your detailed feedback regarding spelling and grammatical issues in the manuscript. We appreciate you pointing out specific examples, and we acknowledge that clarity is critical to ensure the manuscript’s meaning is not distorted. We have now undertaken a thorough manual copyediting of the manuscript beyond automated spell checks, addressing both the examples you noted and other issues throughout the text. The specific corrections you highlighted have been implemented as follows:

• Line 27: Removed the unnecessary word “were” to improve sentence structure.

• Line 44: Corrected the spelling of “formal” (previously missing “L”).

• Line 94: Corrected the misspelling of “deter” (previously written as “dater”).

• Line 146: “Sub-Saharan African” has now been fully spelled out on first mention for clarity.

• Line 313: Removed the redundant word “patient” at the end of the sentence.

• Line 338–339: Revised the sentence for clarity to read: “...had not contacted any other service providers before presenting to the health facility.”

• Line 403 Inserted “who” after “Among those” to complete the sentence grammatically.

We have also carefully reviewed the rest of the manuscript to ensure that similar errors have been corrected and the overall language has been improved for clarity and accuracy.

4. Stigmatising language: Line 164) please mitigate stigmatising language HIV-positive => People/ participants living with HIV; diabetic patients => people/ participants with diabetes mellitus

Response: Thank you for highlighting the use of potentially stigmatising language. We fully agree with the importance of using person-first, respectful terminology in all scientific writing. We have now revised the manuscript to replace phrases such as “HIV-positive” and “diabetic patients” with more appropriate alternatives, including “people living with HIV” and “people with diabetes mellitus.” We have reviewed the entire manuscript to ensure consistent and respectful language is used throughout.

Page 6 Line114-115; Page 8 Line 166-167; Page 29 Line 560; Page 16 Line 335-336

5. Inclusion criteria: Line 189) it strikes me that the ability to comfortable communicate in English or Kiswahili and provide written consent might exclude people with less or no formal education and thus potentially less socio-economic resources/ security - i.e. at higher risk of being affected by catastrophic costs, thus potentially underestimating the true OOP expenditures. Has this been considered and mitigated? It should at least be acknowledged.

Response: Thank you for this important observation. You are absolutely right to highlight the risk of excluding individuals with lower levels of formal education or literacy, which could potentially introduce bias and underestimate out-of-pocket (OOP) expenditures. We would like to clarify that one of the inclusion criteria was the ability to communicate in either Kiswahili or English, depending on the participant’s preference. In Tanzania, Kiswahili is the national language and widely spoken across the population. While most participants chose to communicate in Kiswahili, a few preferred English, and they were not excluded from the study. Furthermore, individuals with low literacy or who were unable to read or write were also not excluded. In such cases, the informed consent process was conducted verbally, with the entire consent form read aloud to the participant. Consent was then obtained using a thumbprint, in the presence of an impartial witness, in accordance with ethical approvals and national research guidelines.

Page 10 Line 198-203

6: Table 1: Line 285/286) the table would benefit from proper lines to better distinguish which sub-themes belong to which theme.

Response: Thank you for this helpful suggestion. We have revised Table 1 to include clearer visual formatting, including horizontal lines and indentation where appropriate, to better distinguish sub-themes from main themes. This improves the readability and clarity of the table, making it easier to follow the thematic structure of the findings.

Page 15 line 295

7. P-values Lines 320-322) “Other variables, such as age, sex, education level, HIV status, smear microscopy results, and treatment regimen, show no significant differences in pre-treatment costs, based on the p-values calculated in the tests (Table 2)” - the sentence refers to p-values in table 2 – which do not exist in said table please rephrase

Response: Thank you for this helpful observation. We acknowledge the oversight and we revised the table accordingly. The sentence has been rephrased to accurately reflect the contents of the updated table, which now includes the p-values for comparisons of pre-treatment costs across the listed variables. Page 17; Line 330-333

8. Figure 1: Line 337) Figure 1 would benefit from clearer labelling of the x axis (what do the numbers 1-4 denote?) and some more description. As the values are percentages of visits in each category, but the number of people in each group likely decreases is the figure not slightly misleading? I would at least expect a number of respondents indicated for each group

Response: Thank you for this helpful comment. We agree that Figure 1 would benefit from clearer labeling and additional contextual information. We have revised the x-axis labels to clarify what the numbers 1–4 represent, and we have updated the figure caption to explicitly describe the figure content. To address the concern regarding potentially misleading interpretation due to decreasing sample sizes across visit categories, we have now included the number of respondents (n) in each group within the figure itself and clarified this in the caption. This will allow readers to better interpret the percentages shown in relation to the underlying group sizes. (Figure 1 attached)

9. Table 3: Line 349) Table 3 shows that the visit time at provider increases for each subsequent visit, not the travel time - the travel time only increases between 2nd and 3rd visit

Response:

Thank you for this helpful observation. We appreciate the reviewer’s attention to detail, which has helped improve the accuracy of our reporting. We agree with the reviewer’s assessment. The text has been revised to accurately reflect the data presented in Table 3. Specifically, we now note that the visit time at each provider increases with each subsequent visit, while travel time remains relatively stable between the first and second visits and only increases between the second and third provider.

Page 18; Line 355-358

9. Discussion: 543/544 I would propose that “worsening condition with potentially worse treatment outcome (e.g. long term-disability) of the affected person and ongoing transmission” should be the first concerns before MDR TB (as long as no TB drugs are use pre-diagnostic, the development of drug resistance is unlikely) – DR TB becomes an issue when Pharmacies/ drug sellers or informal providers provide TB drugs of insufficient strengths and unproven combinations. I would suggest referencing the immediate risks to patients and programme first, as they are touched upon by the respondents already (“I got worse” “I lost all my possessions”) – DR-TB does not seem the main issue in your catchment area, as only 3 out of your 261 participants were diagnosed with/ treated for DR-TB (1%)

Response: Thank you for this valuable suggestion. We agree that emphasizing the immediate risks to patients and the health programme such as worsening of condition, long-term disability, ongoing transmission should precede discussion of MDR-TB in the manuscript. We also acknowledge that in our study area DR TB appears relatively rare (1% of participants), which means other harms may be more urgent or visible to the affected individuals.

Page 29, Line 554-557; Line 595-597

11. COVID-19 Line 599-602: by the very nature of things, it is hard to distinguish COVID-19 from TB just by symptomatic assessment – no education programme with change this. The key is that patients expressed fear of being diagnosed with COVID-19 and then having to face isolation and additional stigma. Therefore the 2 logical conclusions would be to a) ensure that anyone presenting with respiratory symptoms is tested for both TB and COVID-19; b) public health emergency measures are designed in a way that they do not frighten people suffering from other diseases away from accessing services. Policy makers, service providers and the public need the right information and stigma prevention measures.

Response: Thank you for this valuable insight. We agree that symptom overlap between COVID-19 and TB makes it inherently difficult to distinguish the two based on clinical presentation alone, and that no amount of patient education can fully resolve this diagnostic uncertainty. We also acknowledge your important point that fear of a COVID-19 diagnosis and the resulting stigma, isolation, or other consequences was a key barrier to care-seeking during the pandemic. We have revised the manuscript to reflect this nuance more accurately and to highlight the need for integrated testing, appropriate risk communication, and stigma-sensitive public health strategies, as you rightly suggest. Page 32; Line 627-635

---

## [Editor Report · Decision Letter 2]

22 Oct 2025

Implications of pre-diagnosis costs incurred by patients and their families for tuberculosis-related health-seeking behaviours in Mbeya and Songwe regions, Tanzania

PONE-D-24-27964R2

Dear Dr. Stella,

We’re pleased to inform you that your manuscript has been judged scientifically suitable for publication and will be formally accepted for publication once it meets all outstanding technical requirements.

Kind regards,

Bilal Ahmad Rahimi, M.D., D.T.M.&H., M.C.T.P., Ph.D

Academic Editor

PLOS ONE
---

## [Editor Report · Acceptance letter]

PONE-D-24-27964R2

PLOS ONE

Dear Dr. KILIMA,

I'm pleased to inform you that your manuscript has been deemed suitable for publication in PLOS ONE. Congratulations! Your manuscript is now being handed over to our production team.

Kind regards,

on behalf of

Professor Bilal Ahmad Rahimi

Academic Editor

PLOS ONE